# MOTRv3: Release-Fetch Supervision for End-to-End Multi-Object Tracking

## Abstract

Although end-to-end multi-object trackers like MOTR enjoy the merits of simplicity, they suffer from the conflict between detection and association, resulting in unsatisfactory convergence dynamics. While MOTRv2 partly addresses this problem, it demands an additional detector. In this work, we serve as the first to reveal this conflict arises from unfair label assignment between detect and track queries, where detect queries are responsible for recognizing newly appearing targets and track queries are to associate them in following frames. Based on this observation, we propose MOTRv3, which balances the label assignment using the proposed release-fetch supervision strategy. In this strategy, labels are first released for detection and gradually fetched back for association. Besides, another two strategies named pseudo label distillation and track group denoising are designed to further strengthen the supervision for detection and association. Without extra detector during inference, MOTRv3 achieves impressive performance across diverse benchmarks, showing scaling up capability.

## 1 Introduction

Thanks to the broad practical applications like autonomous driving and robotic navigation, multi-object tracking (MOT) is gaining increasing attention from both the research and industry communities (Bewley et al., 2016; Wojke et al., 2017). Early MOT methods mostly adopt the *tracking-by-detection* paradigm, which first recognizes targets using detection networks (Ge et al., 2021; Ren et al., 2015) and then associates them based on appearance similarity (Wang et al., 2020a; Yu et al., 2022a; Zhang et al., 2021) or box Intersection-over-Union (IoU) (Zhang et al., 2022a). Although some of these methods achieve promising performance, all them demand troublesome post-processing operations, e.g., non-maximum suppression (Ren et al., 2015).

In recent years, notable efforts have been paid to remove these post-processing operations (Meinhardt et al., 2021). Among them, MOTR (Zeng et al., 2022) is a milestone, because it unifies the detection and association branches of MOT into a universal Transformer-based architecture and realizes end-to-end tracking without post-processing. Specifically, as shown in Fig. 1a, MOTR first employs detect queries to recognize newly appearing targets like DETR (Carion et al., 2020). When a target is located by a detect query, a track query is generated based on this detect query, and the generated track query is utilized to continuously locate this target in the following frames. Summarily, the detect queries are used for locating newly appearing targets and the track queries are employed to associate these targets in the next frames.

Although the MOTR architecture is concise and straightforward, it suffers from the optimization conflict between detection and association critically, which results in poor detection precision. To alleviate this problem, significant efforts have been paid by some researchers (Cai et al., 2022; Zhang et al., 2022b). For example, as illustrated in Fig. 1b, MOTRv2 employs an independently trained 2D object detector like YOLOX (Ge et al., 2021) to distinguish targets and provide the results to the tracking network. Then, the tracking network can concentrate on association, and thus the conflict is alleviated. Nevertheless, MOTRv2 demands an extra well-trained detector, which makes the tracker not end-to-end any more.

We argue that MOTRv2 does not reveal the essence of the conflict between detection and association in MOTR, and its application is restricted as a well-trained detector is required for inference. In this work, we aim to explore the dark secret of this conflict and provide strategies to tackle it.

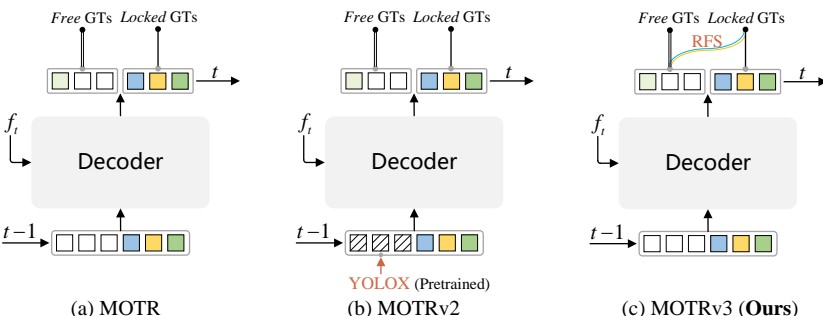

Figure 1: **Comparison among MOTR series.** The main differences in MOTRv2 and MOTRv3 compared with MOTR are marked in red brown. For MOTRv1/v2, *Locked* GTs are the labels that are assigned to track queries and *free* GTs are the ones used to train detect queries.

To this end, we conduct numerous experiments to analyze the training dynamics of MOTR and observe that the activation times of detect queries (the times of detect queries are assigned to the ground-truths) are much smaller compared with the total number of ground-truths. This is because when a detect query matches with a target, the ground-truths of this target in the following frames are fixedly assigned to the track query generated from that detect query, and we call these fixedly assigned ground-truths as locked ground truth (*locked* GT). In this way, the detect query will not receive supervision from this target any more after the first time of assignment. Only the box annotations of newly appearing targets (called *free* GT) are effective for training detect queries. As *free* GT only counts for a small ratio of the total labels, the supervision applying to detect queries is quite limited. This problem causes that the detection part of MOTR is not sufficiently trained.

To tackle this problem, we propose a label assignment strategy named **R**elease-**F**etch **S**upervision (RFS) (see Fig. 1c). It overcomes the restriction of label assignment in MOTR and achieves balanced training between detect and track queries, while keeping the end-to-end spirit. Specifically, in RFS, we release the labels originally assigned only for track queries (*Locked* GT) to detect queries, which means all queries are allowed to compete for the allocation of all GTs. Notably, there are totally 6 decoders and RFS is only applied to the first 5 decoders. The matching strategy of the last decoder in MOTRv3 remains unchanged as MOTR, which ensures track queries can learn to follow the same targets in different frames.

Besides, another two strategies, namely pseudo label distillation (PLD) and track group denoising (TGD), are proposed in this work to further improve the detection and association supervision, respectively. Specifically, PLD uses a well-trained 2D object detector like YOLOX (Ge et al., 2021) or Sparse RCNN (Sun et al., 2021) to produce pseudo labels and apply auxiliary supervision to MOTR. The distribution of pseudo labels provided by the pre-trained detector is diverse, thereby the MOTR detection part obtains more sufficient training. TGD augments track queries into multiple groups and every group consists of the same number of track queries as the original ones. Random noise is added to the reference points of each track group during training. TGD stabilizes the training of the MOTR association part and thus improves the overall tracking performance.

Comprehensively, in this work, we reveal the underlying reason that causes the poor detection performance of MOTR, which previously is simply believed because of the conflict between detection and association. Based on the observation, we propose three strategies that boost the performance of MOTR by a large margin while avoiding the use of an independently trained 2D object detector like MOTRv2. Combining the developed techniques, we propose MOTRv3, which achieves impressive performances across multiple benchmarks including MOT Challenge (Milan et al., 2016; Dendorfer et al., 2020) and DanceTrack (Sun et al., 2022). We hope this work can inspire more researchers about how to improve the end-to-end trackers.

## 2 RELATED WORKS

**Tracking by detection.** Thanks to the fast development of object detection techniques (Ren et al., 2015; Zhou et al., 2019; Ge et al., 2021), existing MOT methods mainly follow the tracking-by-detection (TBD) paradigm (Bewley et al., 2016; Wojke et al., 2017; Bergmann et al., 2019; Peng

et al., 2020; Pang et al., 2020; Wang et al., 2020b), which first uses detectors to locate targets and then associate them to obtain tracklets. According to the association strategy, MOT methods can be further divided into motion-based trackers and appearance-based trackers. Specifically, motion-based trackers (Zhang et al., 2022a; Cao et al., 2022) perform the association step based on motion prediction algorithms, such as Kalman Filter (Bishop et al., 2001) and optical flow (Baker & Matthews, 2004). Some motion-based trackers (Feichtenhofer et al., 2017; Bergmann et al., 2019; Han et al., 2022; Zhou et al., 2020; Sun et al., 2020; Shuai et al., 2021) directly predict the future tracklets or displacements in future frames compared with the current frame. In contrast to the motion-based methods, the appearance-based trackers (Wang et al., 2020a; Zhang et al., 2021; Yu et al., 2022b;a) usually use a Re-ID network or appearance sub-network to extract the appearance representation of targets and match them based on representation similarity.

**End-to-end MOT.** Although the performance of TBD methods is promising, they all demand troublesome post-processing operations, e.g., non-maximum suppression (NMS) (Ren et al., 2015) and box association. Recently, the Transformer architecture (Vaswani et al., 2017) originally designed for natural language processing (NLP) has been applied to computer vision. For instance, DETR (Carion et al., 2020) turns 2D object detection into a set prediction problem and realizes end-to-end detection. Inspired by DETR, MOTR (Zeng et al., 2022) transfers MOT to a sequence prediction problem by representing each tracklet through a track query and dynamically updating track queries during tracking. In this way, the tracking process can be achieved in an end-to-end fashion. However, despite MOTR enjoys the merits of simplicity and elegance, it suffers the limitation of poor detection performance compared to the TBD methods. To improve MOTR, MeMOT (Cai et al., 2022) builds the short-term and long-term memory bank to capture temporal information. LTrack (Yu et al., 2022c) introduces natural language representation obtained by CLIP (Radford et al., 2021) to generalize MOTR to unseen domains. MOTRv2 (Zhang et al., 2022b) incorporate the YOLOX (Ge et al., 2021) detector to generate proposals as object anchors, providing detection prior to MOTR.

## 3 PRELIMINARIES

MOTRv3 is implemented based on MOTR rather than MOTRv2 since it requires an extra 2D object detector, making the tracker not end-to-end. Since not all readers are clear about the design of MOTR, we first elaborate on its architecture in this section. For more details, please refer to the corresponding papers (Zeng et al., 2022; Zhang et al., 2022b). Afterwards, we describe how we reveal the essence resulting in the conflict between detection and association in MOTR.

### 3.1 MOTR PIPELINE

MOTR consists of a CNN backbone, 6 transformer encoder layers, and 6 transformer decoder layers. It realizes end-to-end tracking by applying simple modifications to DETR. Specifically, when a target appears in a video, MOTR employs a detect query to recognize it in the same process as DETR. After recognizing it, MOTR uses a lightweight network block to generate a track query based on this detect query. Then, in the following frames, this track query should continuously locate the positions of this target until it disappears. In a nutshell, the detect queries are utilized to detect newly appeared targets and track queries are for tracking previously detected targets.

For training MOTR, every target in a frame is annotated with a 2D box and an identity. To enable the MOTR detection part to recognize newly appearing targets, the 2D boxes of these new targets are assigned to train detect queries during training. By contrast, if a target exists in previous frames, its GT is used to train the track queries.

### 3.2 A CLOSER LOOK AT LABEL ASSIGNMENT

Although the MOTR architecture is simple and presents promising association accuracy, its detection precision is poor. Previous literature (Zhang et al., 2022b) commonly believes this is due to the conflict between detection and association, but no one reveals where this conflict arises from.

To shed light on this problem, we conduct an in-depth analysis. As suggested in Fig. 2(a), the activation numbers of detect queries with different IDs are limited. We further compare the numbers of 2D box labels that are released to train the detect and track queries (see Fig. 2(c)). It can be observed

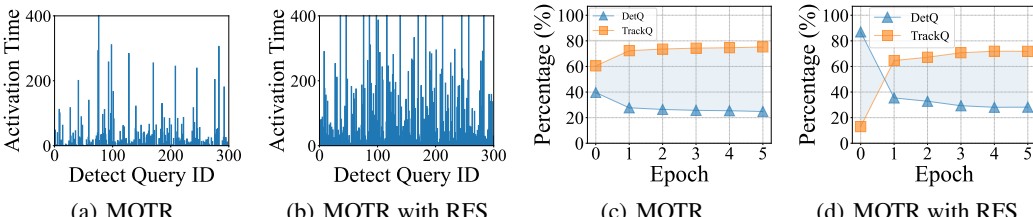

(a) MOTR  (b) MOTR with RFS  (c) MOTR  (d) MOTR with RFS

Figure 2: Fig. (a) and (b) show the activation number of different detect queries with and without the proposed RFS strategy during the training process. Fig. (c) and (d) illustrate the dynamic percentage of ground-truths assigned to the detect queries (DetQ) and track queries (TrackQ) with and without RFS. Note that the experiments are conducted on DanceTrack dataset with overall 5 epochs.

that in the first epoch over 60% labels are used to train the track queries while only 40% are for the detect queries. In the following epochs, the percentage of labels assigned to track queries gradually grows, and the detect queries constantly cannot receive sufficient supervision. These phenomena indicate that the current training paradigm leads to a serious imbalance in the optimization of the detection and association components. In other words, the majority of labels are utilized to supervise the association part, leading to optimal bottleneck of overall model performance, particularly the poor detection performance. To this end, we propose MOTRv3, a pure end-to-end tracking model with a more rationalized supervision method.

## 4 MOTRv3

### 4.1 OVERVIEW

As mentioned before, MOTRv3 is the same as MOTR except the three contributions, i.e., RFS, PLD, and TGD. In this section, we elaborate on the details of them one by one. Among them, RFS conducts one-to-one matching between all GTs and all queries to train the detection capability of the MOTRv3, which is different from MOTR that performs matching between *free* GTs and detect queries. RFS releases the labels originally used for training track queries in MOTR to train the detect queries and gradually fetches them back with the progress of the training process. In PLD, a pre-trained detector is employed to produce more pseudo GTs to train the MOTR detection part more sufficiently. TGD improves the training dynamics stability of the association part by expanding track queries into several groups and then conducting the one-to-one assignment.

### 4.2 RELEASE-FETCH SUPERVISION

Fixedly assigning locked GTs to track queries hinders detect queries getting sufficient supervision. Therefore, we alter this assignment strategy. Specifically, as depicted in Fig. 3, from the $1_{\text{st}}$ to the $5_{\text{th}}$ decoders, the one-to-one matching is performed between all GTs and queries, including both detect queries and track queries, to calculate detection loss. The assignment is dynamic based on matching cost rather than associating locked GTs with fixed track queries. In this way, all detect queries and track queries get abundant supervision. For the $6_{\text{th}}$ decoder, the assignment strategy remains the same as MOTR, which ensures that track queries can learn to follow desired targets.

Compared with the assignment strategy in MOTR, the matching strategy for the $1_{\text{st}}$ to the $5_{\text{th}}$ decoders can be interpreted as the track queries release some locked GTs to detect queries to assist training. Interestingly, with the progress of training, the track queries gradually learn to follow the same targets in various frames, and the number of GTs matched with detect queries decreases continuously. This phenomenon can be understood as after the detect queries get sufficient training, the aforementioned released GTs are fetched back to track queries automatically.

In the following, we formulate RFS in a mathematical form to explain its details. For the $i_{\text{th}}$ frame in a video, assume there are $K$ labels $\hat{y}^i = \{\hat{y}^i_j\}^K_{j=1}$, $M$ detect queries $q^d = \{q^d_j\}^M_{j=1}$, and $N$ track queries $q^t = \{q^t_j\}^N_{j=1}$ (usually $M + N > K$). In MOTRv3, there are two parallel matching strategies, one for detect queries and the other for track queries. In the first one, the labels $\hat{y}^i_d$ of newly appeared targets are assigned to detect queries $q^d$ based on Hungarian matching (Kuhn,

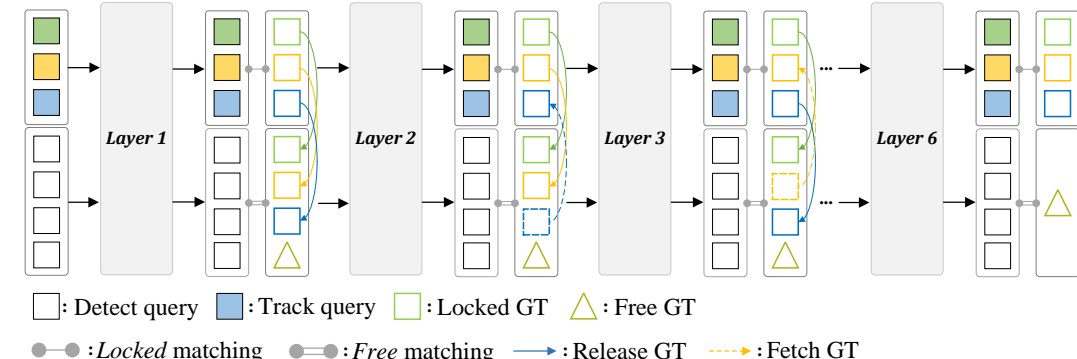

: Detect query    : Track query    : Locked GT    △ : Free GT

●—● : *Locked* matching    ●—● : *Free* matching    ⟶ : Release GT    ⇢ : Fetch GT

Figure 3: **Illustration of Release-fetch supervision (RFS)** in MOTRv3. Notably, RFS is conducted in the first 5 decoder layers and the last layer remains the original supervision.

1955). Mathematically, for the $l_{\text{th}}$ decoder layer ($l = 1, ..., L$), the process:

$$\hat{\sigma}_d^{(i,l)} = \underset{\sigma_d^{(i,l)} \in \mathfrak{S}_d^{(i,l)}}{\arg\min} \sum_{j=1}^{M} \mathcal{L}\left(d_j^{(i,l)}, \hat{y}_{\sigma_d^{(i,l)}(j)}^{(i,l)}\right), \tag{1}$$

where $\mathfrak{S}_d$, $\sigma_d^{(i,l)}$, $d_j^{(i,l)}$, and $\mathcal{L}(\cdot)$ denote the matching space, a sampled matching combination from $\mathfrak{S}_d$, the detection result decoded from the detect query $q_j^d$, and the matching loss, respectively. $\hat{\sigma}_d^{(i,l)}$ represents the optimal matching result.

Notably, the matching space $\mathfrak{S}_d$ is different between the $1_{\text{st}} \sim 5_{\text{th}}$ decoders and the $6_{\text{th}}$ decoder. In the $6_{\text{th}}$ decoder, $\mathfrak{S}_d$ contains all possible matching combinations between $q^d$ and $\hat{y}_d^i$, which means only labels of newly appearing targets can be associated with detect queries for computing loss. Conversely, for the $1_{\text{st}} \sim 5_{\text{th}}$ decoders, $\mathfrak{S}_d$ includes possible matching combinations between all queries ($q^d$ and $q^t$) and all labels ($\hat{y}^i$). In this way, all labels are adopted to train both detect and track queries in the detection loss part and the detection part of the model can obtain more supervision as shown in Fig. 2(b). As illustrated in Fig. 2(d), since $q^t$ cannot precisely follow the locations of targets at the beginning of training, the labels are mostly released to train $q^d$. Then, after $q^t$ gradually be able to correctly recognize the locations of corresponding targets, the labels are fetched back to train $q^t$ automatically.

### 4.3 PSEUDO LABEL DISTILLATION

RFS releases more supervision labels to the detection part by changing the matching strategy. In PLD, we further enhance the supervision applied to the detection part by generating pseudo labels using a previously trained 2D object detector like Sparse RCNN, as shown in Fig. 4a. There are three main benefits of using a pre-trained detector to generate pseudo labels for auxiliary supervision. (i) The pseudo bounding boxes are unbiased representation (diverse and reasonable distribution as shown in Fig. 4a) of objects for high-quality detectors. (ii) Introducing more box annotations would greatly unleash the capacity of object queries by training them with various one-to-one assignment patterns. (iii) The confidence score predicted by detectors can be regarded as a good indicator to represent the quality of bounding boxes, which is embeded with the knowledge of the pre-trained detectors about the quality of predictions. **Notably, the detector is only adopted in training**, which is different from MOTRv2 that still demands this detector in the inference stage.

In PLD, we use the pretrained 2D object detector to generate detection boxes and employ a confidence threshold (such as 0.05) to select precise ones from these boxes. The selected boxes $\hat{y}_e^i$ are used as pseudo labels to train the queries of all 6 decoders. Besides the training process in RFS, we conduct one-to-one matching between all queries ($q^d$ and $q^t$) and $\hat{y}_e^i$ to compute detection loss. In this way, $q^d$ obtains more supervision.

Although the aforementioned process increases the labels for training $q^d$, the problem is that $\hat{y}_e^i$ is often noisy. To alleviate this problem, we propose to reweight the detection part loss based on the detection confidence $c_e$ produced by the 2D object detector. Specifically, if a query matches with a

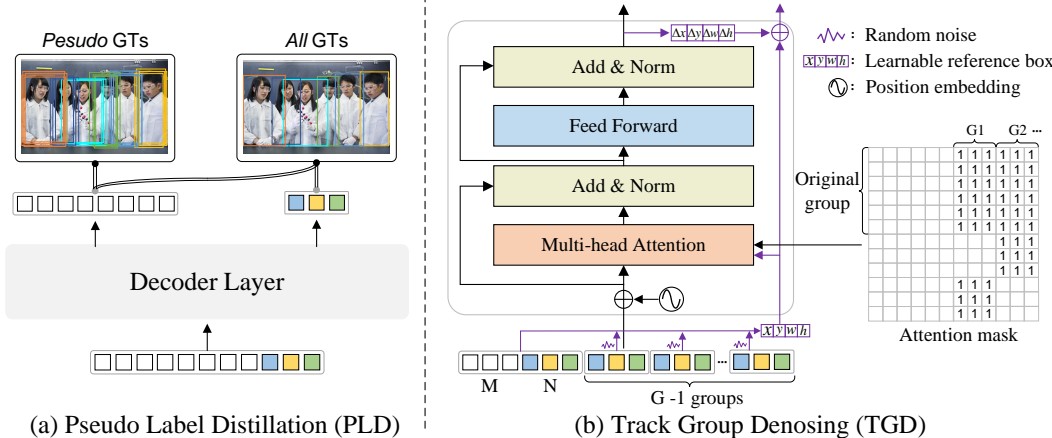

(a) Pseudo Label Distillation (PLD)     (b) Track Group Denosing (TGD)

Figure 4: **Illustration of the PLD (a) and TGD (b) strategy.** We only illustrate the process of one decoder for example, and the other decoders share the same procedures.

label, the loss is multiplied by the confidence value. If no label is matched, the query computes loss with the background class (the same as DETR) and the loss is reweighted by a factor 0.5.

## 4.4 TRACK GROUP DENOSING

The two aforementioned strategies, RFS and PLD, improve the detection capability of MOTR. In this part, we develop a strategy, TGD, to further boost the association performance (see Fig. 4b). Specifically, inspired by Group DETR (Chen et al., 2022), we first augment every track query as a track query group consisting of multiple queries. Notably, the assignment between each track query group and GT is the same as the original track query. By conducting one-to-one matching between track query groups and labels, the track queries obtain more sufficient supervision.

Besides, we note that the tracking performance is influenced by the quality of initial reference points (Zhu et al., 2020) significantly. To boost the robustness of the model, we propose to add random noise to the reference point of every element in a track query group. In this way, the model becomes less dependent on promising initial reference points and the association becomes more robust.

Then, an attention mask is used to prevent information leakage (Li et al., 2022) between the original track query and the augmented queries. Mathematically, we use $\mathbf{A} = [a_{ij}]_{S \times S}$ to denote the attention mask for decoders, where $S = G \cdot N + M$. The value in the attention mask is defined as:

$$a_{ij} = \begin{cases} 1, & \text{if } i < M+N \text{ and } j > M+N; \\ 1, & \text{if } i \geq M+N \text{ and } \lfloor \frac{i-(M+N)}{N} \rfloor \neq \lfloor \frac{j-(M+N)}{N} \rfloor; \\ 0, & \text{otherwise.} \end{cases} \tag{2}$$

where $i$ and $j$ denote the IDs of two queries and $a_{ij}$ defines whether there should exist information communication between these two queries.

## 4.5 LOSS FUNCTION

The entire tracker is optimized with a multi-frame loss function the same as MOTR. The loss function for each frame is formulated as: $\mathcal{L} = \lambda_{cls}\mathcal{L}_{cls} + \lambda_{l_1}\mathcal{L}_{l_1} + \lambda_{giou}\mathcal{L}_{giou}$, where $\mathcal{L}_{cls}$, $\mathcal{L}_{l_1}$, and $\mathcal{L}_{giou}$ are the focal loss (Lin et al., 2017), $L_1$ loss and IoU loss. $\lambda_{cls}$, $\lambda_{l_1}$, $\lambda_{giou}$ are the corresponding hyper-parameters. After expanding the original matching space through our proposed RFS, PLD and TGD strategy, we then calculate the overall clip loss $\mathcal{L}_{clip}$ according to the matching results. Mathematically, it is formulated as:

$$\mathcal{L}_{clip} = \sum_{i=1}^{T} (\mathcal{L}_{\sigma_r^i} + \mathcal{L}_{\sigma_p^i} + \mathcal{L}_{\sigma_g^i})/O_i, \tag{3}$$

where $\sigma_r^i$, $\sigma_p^i$ and $\sigma_g^i$ denote the matching results in the $i^{th}$ frame obtained by RFS, PLD and TGD, respectively. The corresponding $\mathcal{L}$ represents the loss based on the matching results. $T$ is the length of the video clip and $O_i$ is the number of the objects in the $i^{th}$ frame.

## 5 EXPERIMENTS

### 5.1 DATASETS AND METRICS

We conduct extensive experiments on three public datasets, including DanceTrack (Sun et al., 2022), MOT17 (Milan et al., 2016) and MOT20 (Dendorfer et al., 2020), to evaluate the superiority of MOTRv3. In this part, we introduce the adopted datasets and corresponding evaluation metrics.

**DanceTrack** (Sun et al., 2022) is a large-scale multiple object tracking dataset with 100 video sequences in dancing scenarios. The 100 sequences are divided into 40, 25, and 35 sequences for training, validation, and testing, respectively. The targets in DanceTrack are often highly similar in appearance but present various dancing movements. This characteristic causes huge challenge to the association in MOT. In addition, the video sequences in DanceTrack are quite long (52.9 seconds on average for a sequence), which further enhances the tracking difficulty.

**MOT17** (Milan et al., 2016) consists of 14 video sequences. Among them, 7 sequences are for training and the other 7 sequences are used to validate models. These sequences cover various scenarios and weather conditions, which include indoor and outdoor, day and night, etc. The targets in these video sequences are usually pedestrians moving in simple patterns, such as walking straight.

**MOT20** (Dendorfer et al., 2020) presents a heightened level of complexity compared to MOT17. Comprising eight video sequences, it features three bustling environments, with certain frames accommodating over 220 pedestrians concurrently. MOT20 encompasses a wide variety of scenes, ranging from indoor to outdoor settings and spanning various times of day, including nighttime.

**Metrics.** The metrics adopted in the aforementioned datasets include the HOTA (Luiten et al., 2021) and CLEAR-MOT Metrics (Bernardin & Stiefelhagen, 2008). Specifically, HOTA consists of higher order tracking accuracy (HOTA), association accuracy score (AssA), and detection accuracy score (DetA). CLEAR-MOT Metrics include ID F1 score (IDF1), multiple object tracking accuracy (MOTA) and identity switches (IDS).

### 5.2 EXPERIMENTAL SETTINGS

Following MOTR and MOTRv2 (Zeng et al., 2022; Zhang et al., 2022b; Yu et al., 2022c), MOTRv3 is implemented based on Deformable-DETR (Zhu et al., 2020), which is pre-trained on COCO (Lin et al., 2014) and employs ConvNext-Base (Liu et al., 2022) as the vision backbone. During the training process, the batch size is set to 8. For the experiments in DanceTrack and MOT17, each batch is a video clip including 5 frames, which are selected from a video with a random sampling interval between 1 to 10. Following MOTRv2, track queries are generated based on detect queries when the confidences of these detect queries are above the threshold 0.5. Adam (Kingma & Ba, 2014) optimizer is employed and the initial learning rate is set to $2 \times 10^{-4}$.

For the experiments in DanceTrack (Sun et al., 2022), the models are trained for 5 epochs and the learning rate is dropped by a factor of 10 at the $4_{\text{th}}$ epoch. In MOT17 (Milan et al., 2016), we train models for 50 epochs and the learning rate drops at the $40_{\text{th}}$ epoch.

$\lambda_{cls}$, $\lambda_{l1}$ and $\lambda_{giou}$ are set to 2, 5 and 2, respectively. For the implementation of PLD, the auxiliary boxes from pre-trained detectors are obtained in an offline manner. Two common 2D object detectors are adopted, which include YOLOX (Ge et al., 2021) and Sparse RCNN (Sun et al., 2021). The generated 2D box predictions with confidence scores below 0.05 are removed. In the implementation of TGD, we expand the original track query to 4 track query groups.

### 5.3 COMPARISON WITH STATE-OF-THE-ART METHODS

In this part, we compare MOTRv3 with preceding state-of-the-art methods on the three aforementioned MOT benchmarks, *i.e.*, DanceTrack, MOT17 and MOT20. The results on these three benchmarks are reported in Tab. 1-3, respectively. Without bells and whistles, MOTRv3 outperforms all compared methods in the end-to-end fashion.

**DanceTrack.** The results on the DanceTrack test set are presented in Tab. 1. As reported, MOTRv3 outperforms the baseline method MOTR (Zeng et al., 2022) by more than **16** HOTA points on the test set (70.4% vs. 54.2% HOTA). Furthermore, the tracking performance of MOTRv3 is better

Table 1: **Tracking results on DanceTrack `test` set.**

| Method | End to end | HOTA↑ | AssA↑ | DetA↑ | MOTA↑ | IDF1↑ |
|---|---|---|---|---|---|---|
| ***CNN-based*** | | | | | | |
| QDTrack (Pang et al., 2021) | ✗ | 54.2 | 36.8 | 80.1 | 87.7 | 50.4 |
| FairMOT (Zhang et al., 2021) | ✗ | 59.3 | 58.0 | 60.9 | 73.7 | 72.3 |
| CenterTrack (Zhou et al., 2020) | ✗ | 41.8 | 22.6 | 78.1 | 86.8 | 35.7 |
| ByteTrack (Zhang et al., 2022a) | ✗ | 47.7 | 32.1 | 71.0 | 89.6 | 53.9 |
| OC-SORT (Cao et al., 2022) | ✗ | 55.1 | 38.3 | 80.3 | 92.0 | 54.6 |
| ***Transformer-based*** | | | | | | |
| TransTrack (Sun et al., 2020) | ✗ | 45.5 | 27.5 | 75.9 | 88.4 | 45.2 |
| MOTR (Zeng et al., 2022) | ✓ | 54.2 | 40.2 | 73.5 | 79.7 | 51.5 |
| MOTRv2 (Zhang et al., 2022b) | ✗ | 69.9 | 59.0 | 83.0 | 91.9 | 71.7 |
| **MOTRv3 (Ours)** | ✓ | **70.4** | **59.3** | **83.8** | **92.9** | **72.3** |

Table 2: **Tracking results on the MOT17 `test` set.** Notably, MOTRv2 uses extra post-processing operations (Zhang et al., 2022b) for MOT17, and we remove them here for fair comparison. ∗ denotes MOTRv2 without post-processing operations.

| Method | End to end | HOTA↑ | AssA↑ | DetA↑ | MOTA↑ | IDF1↑ | IDS↓ |
|---|---|---|---|---|---|---|---|
| ***CNN-based*** | | | | | | | |
| QDTrack (Pang et al., 2021) | ✗ | 53.9 | 52.7 | 55.6 | 68.7 | 66.3 | 3,378 |
| FairMOT (Zhang et al., 2021) | ✗ | 59.3 | 58.0 | 60.9 | 73.7 | 72.3 | 3,303 |
| CenterTrack (Zhou et al., 2020) | ✗ | 52.2 | 51.0 | 53.8 | 67.8 | 64.7 | 3,039 |
| ByteTrack (Zhang et al., 2022a) | ✗ | 63.1 | 62.0 | 64.5 | 80.3 | 77.3 | 2,196 |
| BoT-Sort (Aharon et al., 2022) | ✗ | 65.0 | - | - | 80.6 | 79.5 | 1,257 |
| ***Transformer-based*** | | | | | | | |
| TransTrack (Sun et al., 2020) | ✗ | 54.1 | 47.9 | 61.6 | 74.5 | 63.9 | 3,663 |
| MOTR (Zeng et al., 2022) | ✓ | 57.8 | 55.7 | 60.3 | 73.4 | 68.6 | 2,439 |
| MOTRv2 (Zhang et al., 2022b) | ✗ | 62.0 | 60.6 | 63.8 | 78.6 | 75.0 | - |
| MOTRv2* (Zhang et al., 2022b) | ✗ | 57.6 | 57.5 | 58.1 | 70.1 | 70.3 | 3,225 |
| MeMOT (Cai et al., 2022) | ✓ | 56.9 | 55.2 | - | 72.5 | 69.0 | 2,724 |
| GTR (Zhou et al., 2022) | ✓ | 59.1 | 57.0 | 61.6 | 75.3 | 71.5 | 2,859 |
| MeMOTR (Gao & Wang, 2023) | ✓ | 58.8 | 58.4 | 59.6 | 72.8 | 71.5 | - |
| **MOTRv3 (Ours)** | ✓ | **60.2** | **58.7** | **62.1** | **75.9** | **72.4** | **2,403** |

than MOTRv2 (70.4% vs. 69.9% HOTA) without using an independent 2D object detector, which is trained on numerous extra 2D object detection data. Meanwhile, MOTRv3 achieves better detection precision than MOTRv2 according to the detection metric MOTA (92.9% vs. 91.9% MOTA), which confirms the effectiveness of the proposed strategies, RFS and PLD. *This marks the first time that the purely end-to-end method surpasses the SOTA tracking schemes that are not end-to-end.*

**MOT17.** The experimental results on the MOT17 benchmark are shown in Tab. 2. Similar to the results in DanceTrack, MOTRv3 outperforms MOTR by a large margin, *i.e.*, 2.4% HOTA and 4.6% IDF1. Moreover, The IDS of MOTRv3 is 36.2% lower than MOTR, which suggests that the obtained trajectories are continuous and robust. Compared with MOTRv2, MOTRv3 also behaves better. Furthermore, we find that the performance of MOTRv2 relies heavily on the adopted post-processing operations. If these operations are removed, the performance of MOTRv2 drops sharply, which is 57.6% HOTA and 70.1% MOTA. By contrast, MOTRv3 does not use any extra post-processing operations and still achieves competitive tracking accuracy. Moreover, MOTRv3 surpasses the latest advanced transformer-based trackers, *e.g.*, GTR and MeMOTR, across all metrics. Additionally, it can be observed that ByteTrack, a CNN-based method, behaves promisingly in MOT17, although it performs inferior to MOTRv3 in DanceTrack. We infer that this is because **the target movement trajectories in MOT17 are simple.** Therefore, the targets in MOT17 can be tracked well by combining a strong 2D object detector like YOLOX and hand-crafted post-processing rules.

**MOT20.** Contrary to MOTR and MOTRv2, we also evaluate MOTRv3 on MOT20 with more complex scenarios and denser pedestrians. As shown in Tab. 3, MOTRv3 outperforms a multitude of transformer-based approaches across multiple metrics in a more challenging benchmark.

## 5.4 ABLATION STUDY

In this part, we perform extensive ablation study experiments using the DanceTrack `validation` set to analyze the effectiveness of various proposed strategies in MOTRv3. The baseline method is MOTR with anchor queries. All models are trained using the DanceTrack training set for 5 epochs.

Table 3: **Tracking results on the MOT20 `test` set.** Mainly report transformer-based methods.

| Method | End to end | HOTA↑ | AssA↑ | DetA↑ | MOTA↑ | IDF1↑ | IDS↓ |
|---|---|---|---|---|---|---|---|
| TransTrack (Sun et al., 2020) | ✗ | 48.9 | 45.2 | - | 65.0 | 59.4 | 3,608 |
| TransCenter (Xu et al., 2021) | ✗ | - | - | - | 67.7 | 76.4 | 1,332 |
| MeMOT (Cai et al., 2022) | ✓ | 54.1 | 55.0 | - | 63.7 | 66.1 | 1,938 |
| **MOTRv3** | ✓ | **60.2** | **61.6** | **59.0** | **72.3** | **74.7** | **911** |

Table 4: **Overall ablation study of the proposed strategies.** The performance of the tracker employing all the developed strategies is highlighted in gray .

| | Components | | | Metrics | | | | | |
|---|---|---|---|---|---|---|---|---|---|
| Method | RFS | PLD | TGD | HOTA↑ | AssA↑ | DetA↑ | MOTA↑ | IDF1↑ | IDS↓ |
| 1 Base | | | | 56.6 | 47.0 | 68.4 | 75.3 | 60.0 | 1,662 |
| 2 | ✓ | | | 60.9 | 49.4 | 75.8 | 85.5 | 63.7 | 1,139 |
| 3 | | ✓ | | 59.2 | 46.5 | 75.5 | 84.9 | 61.7 | 1,284 |
| 4 | | | ✓ | 59.6 | 49.7 | 71.9 | 80.0 | 62.1 | 1,804 |
| 5 | ✓ | ✓ | | 61.7 | 50.0 | 76.3 | 86.0 | 64.8 | 1,350 |
| 6 MOTRv3 | ✓ | ✓ | ✓ | **63.9** | **53.5** | **76.7** | **86.8** | **67.2** | **1,151** |

**Overall ablation study.** In this part, we study the overall influence of the three proposed strategies (RFS, PLD, and TGD) on the MOTRv3 performance. The results are reported in Tab. 4. According to the results, all these three strategies boost the tracking performance significantly. Among these strategies, both RFS (row #2) and PLD (row #3) enhance the tracking precision by a large margin. Specifically, RFS improves the MOTA score by $10.2\%$ and DetA score by $7.4\%$. PLD boosts the MOTA score by $9.6\%$ and DetA score by $7.1\%$. The results indicate that both fair assignment strategy and aux supervision improve the detection capability of MOTR quite effectively. Additionally, combing them further improves the tracking performance by a large margin (row #5). This is because RFS guarantees that a proper ratio of labels is released to train the detect queries, and PLD helps generate more detection labels. Combining them enables the MOTR detection part to be sufficiently trained. Moreover, it can be observed that TGD improves the AssA score by $2.7\%$ and IDF1 score by $2.1\%$. This observation indicates that the representing ability of track queries is improved, and thus the produced trajectories become more robust.

Incorporating all these strategies, MOTRv3 (row 6) outperforms the baseline (row 1) by $7.3\%$ on HOTA and $11.5\%$ on MOTA. Summarily, the experimental results demonstrate that the proposed strategies can address the conflict between detection and association existing in end-to-end trackers effectively and MOTRv3 is an efficient end-to-end tracker.

**PLD.** PLD is responsible for producing more training labels. In this part, we study how different pseudo label generation strategies affect tracking performance. we compare the performance among different settings (Sparse RCNN, GT, and GT + noises) of PLD without using other proposed strategies. The results are presented in Tab. 5. Three observations can be drawn. **(i)** Using GT boxes as pseudo labels can also improve the performance. It means both pre-trained detectors and GT are optional for PLD though the improvement of using GT is not as significant as the pseudo-labels (see line 1, 2, and 4). **(ii)** When adding random noises (s denotes the noise scale) to the GT, the performance drops a lot (see line 4, 5 and 6). We analyze the reason for this phenomenon is that adding random noises to GT can not generate reasonable bounding boxes like pre-trained detector. The optimal noise scale is not easy to search, which is harmful to the learning of detection part. **(iii)** We also compute the aux loss of PLD with (w) and without (w/o) using confidence score as weights (see line 2 and 3). The result shows that confidence score is an important part for PLD as it is a good indicator to represent the quality of the bounding boxes.

**TGD.** In this experiment, we study how the track group number affects performance and the influence of noise added to the track query reference points. The results are reported in Tab. 6. As shown in the $1_{st} \sim 4_{th}$ rows of results, augmenting every query into a group improves the performance significantly and setting the query number to 4 results in the best result. Augmenting a query into too many or too few queries both harm the final tracking performance. Besides, adding noise to reference points also boosts the tracking precision significantly, which is given in the $5_{th}$ row. The results suggest that the developed TGD strategy enhances the association accuracy of MOTR significantly, which we believe is because the stability of the training process is improved.

Table 5: **The tracking results of using different pseudo label generation strategies.** S-RCNN is the 2D detector Sparse RCNN and $s$ denotes the noise scale.

| | Pseudo labels | HOTA↑ | AssA↑ | DetA↑ | MOTA↑ | IDF1↑ | IDS↓ |
|---|---|---|---|---|---|---|---|
| 1 | None | 56.6 | 47.0 | 68.4 | 75.3 | 60.0 | 1,662 |
| 2 | S-RCNN (Sun et al., 2021) | **59.2** | **46.5** | **75.5** | **84.9** | **61.7** | **1,284** |
| 3 | S-RCNN w/o score | 57.8 | 45.5 | 73.6 | 83.1 | 59.5 | 1,476 |
| 4 | GT | 58.5 | 45.8 | 74.9 | 84.4 | 59.8 | 1,692 |
| 5 | GT + Noises (s=0.01) | 56.1 | 42.9 | 73.7 | 82.5 | 56.7 | 2,243 |
| 6 | GT + Noises (s = 0.05) | 52.5 | 45.5 | 60.7 | 67.2 | 56.1 | 1,573 |

Table 6: **Ablation Study on how TGD affects the tracking performance.**

| Method | Group Num | HOTA↑ | AssA↑ | DetA↑ | MOTA↑ | IDF1↑ | IDS↓ |
|---|---|---|---|---|---|---|---|
| base | | 61.7 | 50.0 | 76.3 | 86.0 | 64.8 | 1,350 |
| + TG | 3 | 63.2 | 52.4 | 76.6 | 86.1 | 65.9 | 1,116 |
| | 4 | 63.7 | 53.3 | **76.8** | 86.7 | 66.6 | **1,027** |
| | 5 | 63.3 | 52.6 | 76.6 | 86.3 | 66.5 | 1,106 |
| + RN | 4 | **63.9** | **53.5** | 76.7 | **86.8** | **67.2** | 1,151 |

## 5.5 SCALING UP MOTRv3

**Model Scaling Up.** In this part, we mainly study how scaling up backbones affects the tracking performance of MOTRv2 and MOTRv3. Specifically, we replace the original ResNet-50 backbone of them with ConvNeXt-tiny, ConvNeXt-small, and ConvNeXt-base, respectively. The results are illustrated as Fig. 5(a). It can be observed that the performance of MOTRv3 is continuously boosted with the scaling up of backbones. However, scaling up the backbone harms the tracking precision of MOTRv2. We speculate that this is because MOTRv2 needs an extra detector and is not end-to-end, thereby only replacing the backbone of the association part does not improve the overall tracking performance. By contrast, MOTRv3 is fully end-to-end and thus enjoys the benefits from scaling up the model. This represents a significant advantage of MOTRv3 over its predecessors.

**Data Scaling Up.** The primary advantage of pure end-to-end models is their ability to enhance performance by continuously increasing data. However, for pedestrian tracking datasets like MOT17, the limited data scale currently prevents pure end-to-end models from surpassing rule-based traditional detection-based tracking approaches. To validate this assertion, we further augmented MOTRv3 training with MOT20 and Sompt22(Simsek et al., 2022) data based on MOT17. As shown in Fig. 5(b), with the addition of more data,

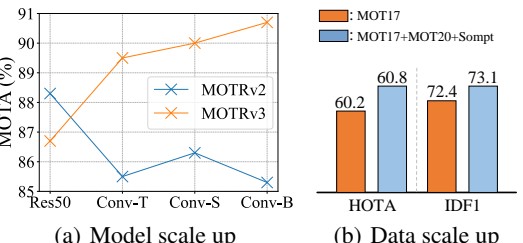

(a) Model scale up          (b) Data scale up

Figure 5: **Scaling up experiments.** We conducted scale-up experiments separately in two aspects: (a) model size and (b) data scale.

the tracking performance of MOTRv3 achieves stable growth. This indicates that our approach will gain further advantages on a larger scale of data.

## 6 LIMITATION AND CONCLUSION

In this work, we reveal the real reason causing the conflict between detection and association in MOTR, which results in the poor detection performance. Based on this observation, we propose RFS, which improves the detection and overall tracking performances by a large margin. However, while RFS helps mitigate this conflict in terms of supervision, the trade-off between detection and association remains unresolved. How to disentangle two sub-tasks still deserves further study. Besides, we have proposed two another strategies, PLD and TGD, to further improve the detection and query parts of MOTR. Combining all the three strategies, the developed tracker, MOTRv3, has achieved impressive performances across multiple benchmarks. We hope this work can inspire more solid works about end-to-end MOT in the future.

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

## A    APPENDIX

In this supplementary material, we provide more details of MOTRv3 due to the 10-pages limitation on paper length. Specifically, Section B elaborates on the auxiliary 2D object detector employed in the pseudo label generation strategy. Section C provides additional experiments to analyze the characteristics of MOTRv3.

## B    AUXILIARY DETECTORS USED IN PSEUDO LABEL GENERATION

In this work, we mainly use YOLOX (Ge et al., 2021) and Sparse RCNN (Sun et al., 2021) detectors to generate pseudo labels.

**YOLOX.** We employ the YOLOX detector that the model weights are from ByteTrack (Zhang et al., 2022a) and DanceTrack (Sun et al., 2022). The hyper-parameters and data augmentation techniques, including Mosaic (Bochkovskiy et al., 2020) and Mixup, remain consistent with ByteTrack. YOLOX-X (Ge et al., 2021) is adopted as the backbone. For the results on MOT17, the model is trained for 80 epochs combining the data from MOT17, Crowdhuman, Cityperson, and ETHZ datasets. Regarding DanceTrack, we directly used the YOLOX weight provided by the DanceTrack official GitHub repository[1].

**Sparse RCNN.** We utilize the original Sparse RCNN implemented in the official repository[2], with the ResNet-50 backbone (He et al., 2016) initialized from a COCO-pretrained model. The number of learnable anchors is set to 500. To train on the MOT17 dataset, we initially train Sparse RCNN on Crowdhuman for 50 epochs. Subsequently, we further fine-tune it on MOT17 for additional 30 epochs. Similarly, for the process on DanceTrack, we also first pre-train Sparse RCNN on Crowdhuman for 50 epochs, and then fine-tune it on DanceTrack for 20 epochs.

## C    ADDITIONAL EXPERIMENTS.

**Ablation study on RFS.** By applying RFS to MOTR, we allow detect queries and track queries to compete for the supervision labels fairly in the first 5 decoders. In this way, the detect queries of MOTRv3 obtain more sufficient supervision compared with MOTR. Nevertheless, RFS could result in inconsistent learned label assignment patterns between the first 5 decoders and the last decoder due to different assignment strategies during training, which may be harmful to the final tracking performance. In this part, we study this issue by visualizing the diversity between the first 5 decoder layers and the last decoder layer during training.

Specifically, for a detect query, if the matched label is different between the last decoder layer and one of the 5 decoder layers, we call the label assignment is misaligned. In this experiment, we count the percentages of misaligned labels relative to the total labels for trackers using and without using RFS. The misalignment percentages of two trackers in various epochs are visualized in Fig. 6. The graph clearly indicates that the usage of RFS amplifies the misalignment of label matching during the initial training epochs, but over time, the percentages gradually decrease and eventually reach the same level as those without RFS. This observation suggests that the

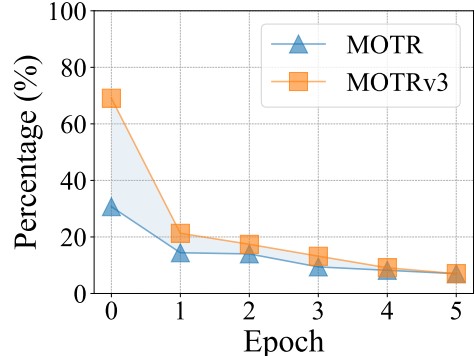

Figure 6: **Matching results diversity** comparsion between MOTR and MOTRv3.

high matching diversity introduced by RFS in the early training stage does not hinder the convergence of the label matching process. In fact, the increased matching diversity allows more queries to participate in the learning process, which ultimately benefits the detection part.

---

[1] https://github.com/DanceTrack/DanceTrack
[2] https://github.com/PeizeSun/SparseR-CNN

Table 7: **The tracking results of using different pseudo label generation detectors.**

| Pretrained detector | HOTA↑ | AssA↑ | DetA↑ | MOTA↑ | IDF1↑ | IDS↓ |
|---|---|---|---|---|---|---|
| YOLOX (Ge et al., 2021) | 61.6 | 49.8 | 76.4 | 86.1 | 63.4 | 1,408 |
| Sparse RCNN (Sun et al., 2021) | **61.7** | 50.0 | 76.3 | 86.0 | **64.8** | **1,350** |
| Parallel (YOLOX & Sparse RCNN) | 60.4 | 47.4 | **77.1** | **86.7** | 62.5 | 1,551 |

Table 8: **Comparisions between TGD with (w) and without (w/o) attention mask**.

| Adapter | HOTA↑ | AssA↑ | MOTA↑ | IDF1↑ | IDS↓ |
|---|---|---|---|---|---|
| w | **63.9** | **53.5** | **86.8** | **67.2** | **1,151** |
| w/o | 63.0 | 53.2 | 84.5 | 66.3 | 1,298 |

Table 9: **Inference speed comparison** on DanceTrack `test` set among MOTR series.

| Method | Backbone | HOTA↑ | MOTA↑ | IDF1↑ | FPS↑ |
|---|---|---|---|---|---|
| MOTR | ResNet-50 | 54.2 | 79.7 | 51.5 | 9.5 |
| MOTRv2 | ResNet-50 | 69.9 | 91.9 | 71.7 | 6.9 |
| MOTRv3 | ConvNeXt-B | **70.4** | **92.9** | **72.3** | **9.8** |

**Ablation of the selected detectors in PLD.** In this part, we study how different pseudo label generation detectors affect tracking performance. Specifically, we compare 3 strategies, i.e., generating pseudo labels by YOLOX or Sparse RCNN, and combining pseudo labels (concat or parallel) from YOLOX and Sparse RCNN. The results are presented in Tab. 7. We can observe employing any one of YOLOX and Sparse RCNN leads to promising performance improvement and combining them further improves the detection performance. However, the association performance tends to decrease when combining in parallel. We speculate that this is because the distributions of boxes generated by them are different and this issue confuses the learning of association during training.

**Effect of Attention Mask in TGD.** As introduced in the Section 4.4 of Manuscript, TGD adopts the attention mask to prevent information leakage between the original track query and the augmented queries. In this part, we design an ablation study about attention mask in Tab. 8. We can observe the performance without designed attention mask has a certain degree of performance decline. The results prove that it is effective to prevent information leakage between the original track query and the track group queries through attention mask.

**Analysis on the inference speed.** As mentioned in the main paper, our proposed strategies, namely RFS, PLD, and TGD, are exclusively employed during training and do not introduce any additional network blocks. Consequently, the inference speed of MOTRv3 remains competitive. As depicted in Tab. C, we compare the inference speeds of MOTR(Zeng et al., 2022), MOTRv2(Zhang et al., 2022b), and MOTRv3 on the DanceTrack `test` set. It can be observed that our MOTRv3, with the larger ConvNext-Base(Liu et al., 2022) backbone achieves superior performance while still maintaining a competitive inference speed (9.8 FPS).

# D ADDITIONAL DISCUSSIONS.

**One-to-One or One-to-Many Assignment in RFS?** In fact, we still employ the original one-to-one assignment strategy in RFS. The core contribution of RFS is to improve the label matching space between the detection and association parts, rather than the assignment method itself. Since MOT requires one-to-one corresponding between track queries and targets for trajectory association, one-to-many assignment is not suitable. Our focus is on the issue of balancing label assignment between the detection and association, rather than label assignment strategy for detection.

