# OpenReview forum: "MOTRv3: Release-Fetch Supervision for End-to-End Multi-Object Tracking"
_ICLR.cc/2025/Conference — ICLR 2025 Conference Withdrawn Submission_

### Official Review · Reviewer_S9H1 · 2024-11-02

**Soundness:** 2
**Presentation:** 3
**Contribution:** 2
**Rating:** 5
**Confidence:** 3

**Summary:**

This paper proposes a novel end-to-end MOT tracker MOTRv3 by introducing a Release-Fetch Supervision (RFS) strategy. This approach enables a dynamic labeling strategy to optimize the detector and later reassigned to the track queries during training. Also, the authors propose Pseudo Label Distillation (PLD), leveraging pre-trained detector to provide pseudo labels and adjusting the loss. For optimizing the association part, a Track Group Denoising (TGD) strategy is to augment e each track query. By integrating above three designs, MOTRv3 outperforms its baseline and achieves SOTA performance among end-to-end MOT methods across benchmarks.

**Strengths:**

1. The idea to alleviate the label imbalance between the optimization of detection and association subtasks in MOT is sound and appealing.
2. The paper is well-organized with theoretical analysis, motivation, and methodology.  Also, comprehensive experimental evaluations and comparisons are provided.
3. The proposed RFS strategy is effective with sufficient explanation. And the good overall performance of the proposed MOTRv3 is inspiring for end-to-end MOT.

**Weaknesses:**

1. For evaluation results on DanceTrack, the performance improvement is very limited compared to MOTRv2.
2. The experiments are not fair and unconvining since MOTRV3 applies the ConvNext-Base as backbone, while others uses ResNet50.
3. The autohrs should further discuss the label assignment of previous object detection and tracking, such as:
[1] Group DETR: Fast DETR Training with Group-Wise One-to-Many Assignment, ICCV 2023
[2] DAB-DETR: Dynamic Anchor Boxes are Better Queries for DETR, ICLR 2022
[3] Probabilistic Assignment with Decoupled IoU Prediction for Visual Tracking, TCSVT2024
[4] Towards Efficient Training with Negative Samples in Visual Tracking, ArXiv 2023
4. More visual tracking results and comparisons are expected.

**Questions:**

1. More detailed explanation of TGD would help the readers understand the proposed method better. For example, what specific augmentation techniques are used to construct query group from a query.
2. What's the main difference in terms of impact on tracking performance between directly integrating YOLO detector (as in MOTRv2) and using PLD proposed in this paper.

---

### Official Review · Reviewer_hnTw · 2024-11-02

**Soundness:** 2
**Presentation:** 2
**Contribution:** 2
**Rating:** 3
**Confidence:** 5

**Summary:**

The paper addresses the problem of poor detection performance of MOTR models that is because of the racing in using ground truth labels between the detector and the tracker components. Three components related to ground truth augmentation and distillation are proposed to address the introduced problem. Performance is improved by a marginal degree.

**Strengths:**

- The finding about the detection/association conflict provided in Fig. 1 is somehow interesting.

**Weaknesses:**

- The novelty is relatively limited and incremental:
   + The improvement can be simply interpreted as adding a lot more pseudo-GT from the TGD components into both detection and association stages from the 1st to 5th decoders, named RFS.
   + The TGD component is changing the assignment to track query group and GT.

- The writing is not clear as many technical details are explained in plain texts rather than algorithms or equations, which compromises readability and reproducibility.

- The numerical improvements are subtle, only from 1 - 3% on main metrics.

- No qualitative comparison showing superior or impressive outperformance is given.

**Questions:**

Please see Weaknesses.

---

### Official Review · Reviewer_kDKQ · 2024-11-04

**Soundness:** 2
**Presentation:** 2
**Contribution:** 2
**Rating:** 3
**Confidence:** 5

**Summary:**

This paper addresses the challenge of resolving conflicts between detection and association in end-to-end multi-object trackers, specifically in MOTR and MOTRv2. The authors identify unfair label assignment between detect and track queries as a key issue. To address this, MOTRv3 introduces three main strategies: release-fetch supervision, pseudo label distillation, and track group denoising. These methods enable MOTRv3 to perform well on benchmarks like MOT17 and DanceTrack, achieving results without requiring an external detection network during inference.

**Strengths:**

Compared to MOTRv2, MOTRv3 returns to a fully end-to-end design, eliminating the dependence on external detection models. This improves the model’s integration and inference efficiency, making it suitable for real-time applications.

**Weaknesses:**

1. Neither v1 nor v2 utilized pseudo label distillation. In v3, Pseudo Label Distillation (PLD) enhances the training of detection queries by using high-quality pseudo labels, further improving detection accuracy. While effective, this approach does not alter the fundamental framework.
2. v3 introduces Track Group Denoising (TGD), which enhances the stability of tracking queries, making the model more robust in dynamic scenes and reducing issues like ID switching. TGD can be seen as an extension of data augmentation at the tracking query level. Its innovation still feels limited.
3. While RFS helps balance supervision signals between detection and tracking queries during training, its core innovation lies in label scheduling rather than in any fundamental changes to the model structure or training process.

**Questions:**

Could you elaborate on why the tracking-by-detection framework continues to outperform the MOTR family?

---

### Official Review · Reviewer_d8DH · 2024-11-07

**Soundness:** 3
**Presentation:** 3
**Contribution:** 2
**Rating:** 5
**Confidence:** 5

**Summary:**

This paper tackles the end-to-end multiple object tracking with transformers. It observes the issue of imbalanced distribution of the detection and tracking labels assignment. The majority of the queries are assigned to the track queries causing the detection queries are not be sufficiently supervised during training. The authors propose release-fetch supervision (RFS) to increase the proportion of detection queries.  Besides RFS, the paper also uses the pseudo labels from other strong detectors to further boost the detection performance of the proposed tracker. The authors also explore augmentation such as adding noise to the reference point and creating track query groups. The experiments demonstrate slightly better performance than MOTRv2 without relying on external detectors on DanceTrack but get worse performance on MOT17 if compared to the original MOTRv2.

**Strengths:**

1. The paper is well-written and clearly organized.

2. The observation regarding the disproportional assignment of track and detection queries is insightful, with a clear analysis that adds valuable understanding for the MOT community.

3. The proposed Release-Fetch Supervision (RFS) method is simple yet effective. Due to its simplicity, it could potentially benefit various Transformer-based MOT methods.

**Weaknesses:**

1. Limited Scenarios for RFS Technique: The need for RFS is primarily in scenarios where there is more video data than image data for training, leading to a disproportionate assignment of track and detection queries. However, in most cases, such as in large-scale [1] and open-vocabulary MOT [2] tasks, the opposite is true, with more image detection data than video tracking data. In these cases, joint training with both image and tracking data is a common practice, providing sufficient supervision for detection queries, contrary to the paper’s analysis. It would be helpful if the authors could at least discuss these more common scenarios.

2.  Besides the proposed RFS technique, other contributions, such as Pseudo Label Distillation (PLD) and Track Grouping Distillation (TGD), have been explored in prior works [3,4]. [3] also utilizes pseudo-detection labels to improve the detector training. [4] also, adds noise to the GT and learning the denoise process in the decoder.  A short discussion of those similar ideas would be beneficial.

3. There is no comparison with simpler alternatives to RFS. A straightforward alternative would be to train detection queries jointly on image detection datasets along with video tracking datasets. For example, detection queries could be trained only on images when using image datasets. I would recommend an ablation study comparing RFS to this joint training approach to provide concrete evidence of RFS’s effectiveness relative to existing methods.

4. The paper uses the first five decoders to train with all queries, while the last one trains separately on detection and tracking queries. An ablation study could clarify whether alternative arrangements, such as using the first decoder for track queries and the remaining five for all queries, would impact performance. To deepen the analysis, I suggest testing multiple configurations (e.g., using the first 1, 3, or 5 decoders for mixed queries) and reporting on their effect on performance metrics. This would provide a more comprehensive understanding of the optimal configuration.

[1] Li, Siyuan, et al. "Tracking every thing in the wild." European Conference on Computer Vision. Cham: Springer Nature Switzerland, 2022.
[2] Li, Siyuan, et al. "Ovtrack: Open-vocabulary multiple object tracking." Proceedings of the IEEE/CVF conference on computer vision and pattern recognition. 2023.
[3] Wu, Di, et al. "Spatial self-distillation for object detection with inaccurate bounding boxes." Proceedings of the IEEE/CVF International Conference on Computer Vision. 2023.
[4] Zhang, Hao, et al. "Dino: Detr with improved denoising anchor boxes for end-to-end object detection." arXiv preprint arXiv:2203.03605 (2022).​

**Questions:**

1. Will the code be publicly available?
2.  Why does denoising only happen with track queries instead of both detection and track queries?

---

### Note · Authors · 2024-11-14

I have read and agree with the venue's withdrawal policy on behalf of myself and my co-authors.